# SAMPLE-EFFICIENT AUTOMATED DEEP REINFORCEMENT LEARNING

**Jörg K.H. Franke[1], Gregor Köhler[2], André Biedenkapp[1] & Frank Hutter[1,3]**
[1]Department of Computer Science, University of Freiburg, Germany
[2]German Cancer Research Center, Heidelberg, Germany
[3]Bosch Center for Artificial Intelligence, Renningen, Germany
`frankej@cs.uni-freiburg.de`

## ABSTRACT

Despite significant progress in challenging problems across various domains, applying state-of-the-art deep reinforcement learning (RL) algorithms remains challenging due to their sensitivity to the choice of hyperparameters. This sensitivity can partly be attributed to the non-stationarity of the RL problem, potentially requiring different hyperparameter settings at various stages of the learning process. Additionally, in the RL setting, hyperparameter optimization (HPO) requires a large number of environment interactions, hindering the transfer of the successes in RL to real-world applications. In this work, we tackle the issues of sample-efficient and dynamic HPO in RL. We propose a population-based automated RL (AutoRL) framework to meta-optimize arbitrary off-policy RL algorithms. In this framework, we optimize the hyperparameters and also the neural architecture while simultaneously training the agent. By sharing the collected experience across the population, we substantially increase the sample efficiency of the meta-optimization. We demonstrate the capabilities of our sample-efficient AutoRL approach in a case study with the popular TD3 algorithm in the MuJoCo benchmark suite, where we reduce the number of environment interactions needed for meta-optimization by up to an order of magnitude compared to population-based training.

## 1 INTRODUCTION

Deep reinforcement learning (RL) algorithms are often sensitive to the choice of internal hyperparameters (Jaderberg et al., 2017; Mahmood et al., 2018), and the hyperparameters of the neural network architecture (Islam et al., 2017; Henderson et al., 2018), hindering them from being applied out-of-the-box to new environments. Tuning hyperparameters of RL algorithms can quickly become very expensive, both in terms of high computational costs and a large number of required environment interactions. Especially in real-world applications, sample efficiency is crucial (Lee et al., 2019).

Hyperparameter optimization (HPO; Snoek et al., 2012; Feurer & Hutter, 2019) approaches often treat the algorithm under optimization as a black-box, which in the setting of RL requires a full training run every time a configuration is evaluated. This leads to a suboptimal sample efficiency in terms of environment interactions. Another pitfall for HPO is the non-stationarity of the RL problem. Hyperparameter settings optimal at the beginning of the learning phase can become unfavorable or even harmful in later stages (François-Lavet et al., 2015). This issue can be addressed through dynamic configuration, either through self adaptation (Tokic & Palm, 2011; François-Lavet et al., 2015; Tokic, 2010) or through external adaptation as in population-based training (PBT; Jaderberg et al., 2017). However, current dynamic configuration approaches substantially increase the number of environment interactions. Furthermore, this prior work does not consider adapting the architecture.

In this work, we introduce a simple meta-optimization framework for *Sample-Efficient Automated RL (SEARL)* to address all three challenges: sample-efficient HPO, dynamic configuration, and the dynamic modification of the neural architecture. The foundation of our approach is a joint optimization of an off-policy RL agent and its hyperparameters using an evolutionary approach. To reduce the amount of required environment interactions, we use a shared replay memory across the population of different RL agents. This allows agents to learn better policies due to the diverse

collection of experience and enables us to perform AutoRL at practically the same amount of environment interactions as training a single configuration. Further, SEARL preserves the benefits of dynamic configuration present in PBT to enable online HPO and discovers hyperparameter schedules rather than a single static configuration. Our approach uses evolvable neural networks that preserve trained network parameters while adapting their architecture. We emphasize that SEARL is simple to use and allows efficient AutoRL for any off-policy deep RL algorithm.

In a case study optimizing the popular TD3 algorithm (Fujimoto et al., 2018) in the MuJoCo benchmark suite we demonstrate the benefits of our framework and provide extensive ablation and analytic experiments. We show a $10\times$ improvement in sample efficiency of the meta-optimization compared to random search and PBT. We also demonstrate the generalization capabilities of our approach by meta-optimizing the established DQN (Mnih et al., 2015) algorithm for the Atari benchmark. We provide an open-source implementation of SEARL.[1] Our contributions are:

- We introduce an AutoRL framework for off-policy RL which enables: (i) Sample-efficient HPO while training a population of RL agents using a shared replay memory. (ii) Dynamic optimization of hyperparameters to adjust to different training phases; (iii) Online neural architecture search in the context of gradient-based deep RL;

- We propose a fair evaluation protocol to compare AutoRL and HPO in RL, taking into account the actual cost in terms of environment interactions.

- We demonstrate the benefits of SEARL in a case study, reducing the number of environment interactions by up to an order of magnitude.

## 2 RELATED WORK

**Advanced experience collection:** Evolutionary RL (ERL) introduced by Khadka & Tumer (2018) and successors PDERL (Bodnar et al., 2020) and CERL (Khadka et al., 2019) combine Actor-Critic RL algorithms with genetic algorithms to evolve a small population of agents. This line of work mutates policies to increase the diversity of collected sample trajectories. The experience is stored in a shared replay memory and used to train an Actor-Critic learner with fixed network architectures using DDPG/TD3 while periodically adding the trained actor to a separate population of evolved actors. CERL extends this approach by using a whole population of learners with varying discount rates. However, this line of work aims to increase a single configuration's performance, while our work optimizes hyperparameters and the neural architecture while training multiple agents. SEARL also benefits from a diverse set of mutated actors collecting experience in a shared replay memory. Schmitt et al. (2019) mix on-policy experience with shared experiences across concurrent hyperparameter sweeps to take advantage of parallel exploration. However, this work neither tackles dynamic configuration schedules nor architecture adaptation.

**ApeX/IMPALA:** Resource utilization in the RL setting can be improved using multiple actors in a distributed setup and decoupling the learner from the actor. Horgan et al. (2018) extends a prioritized replay memory to a distributed setting (Ape-X) to scale experience collection for a replay memory used by a single trainer. In IMPALA (Espeholt et al., 2018), multiple rollout actors asynchronously send their collected trajectories to a central learner through a queue. To correct the policy lag that this distributed setup introduces, IMPALA leverages the proposed V-trace algorithm for the central learner. These works aim at collecting large amounts of experience to benefit the learner, but they do not explore the space of hyperparameter configurations. In contrast, the presented work aims to reduce the number of environment interactions to perform efficient AutoRL.

**Neural architecture search with Reinforcement Learining:** The work of Zoph & Le (2016) on RL for neural architecture search (NAS) is an interesting counterpart to our work on the intersection of RL and NAS. Zoph & Le (2016) employ RL for NAS to search for better performing architectures, whereas we employ NAS for RL to make use of better network architectures.

**AutoRL:** Within the framework of AutoRL, the joint hyperparameter optimization and architecture search problem is addressed as a two-stage optimization problem in Chiang et al. (2019), first shaping the reward function and optimizing for the network architecture afterward. Similarly, Runge et al. (2019) propose to jointly optimize algorithm hyperparameters and network architectures by searching

---

[1]Please find the source code on GitHub: github.com/automl/SEARL

over the joint space. However, they treat the RL training as a black-box and do not focus on online optimization nor sample-efficiency. In contrast to black-box optimization, we jointly train the agent and dynamically optimize hyperparameters. Faust et al. (2019) uses an evolutionary approach to optimize a parametrized reward function based on which fixed network topologies are trained using standard RL algorithms, treating the RL algorithm together with a sampled reward function as a black-box optimizer. In this work, we do not use parametrized reward functions, but instead, directly optimize the environment reward. The main difference to this line of work is sample efficiency: While they train and evaluate thousands of configurations from scratch, we dynamically adapt the architecture and RL-algorithm hyperparameters online, thereby drastically reducing the total amount of interactions required for the algorithm to achieve good performance on a given task. We propose an evaluation protocol taking into account the aspect of sample-efficiency in RL in section 4.3.

**Self-Tuning Actor-Critic:** Zahavy et al. (2020) propose to meta-optimize a subset of differentiable hyperparameters in an outer loop using metagradients. This however, does not extend to non-differentiable hyperparameters and thus does not allow for online tuning of e.g. the network architecture. As a results, such hyperparameters can not be meta-optimized in their framework.

**HOOF:** Paul et al. (2019) proposes sample-efficient hyperparameter tuning for policy gradient methods by greedily maximizing the value of a set of candidate policies at each iteration. In contrast to our work, HOOF performs HPO for on-policy algorithms that do not achieve comparable performance on continuous control tasks while requiring more interactions and not considering architecture optimization.

**Population-Based Training (PBT):** PBT (Jaderberg et al., 2017) is a widely used dynamic and asynchronous optimization algorithm. This approach adapts a population of different hyperparameter settings online and in parallel during training, periodically replacing inferior members of the population with more promising members. Similarly to SEARL, PBT can jointly optimize the RL agent and its hyperparameters online, making it the most closely related work. Recent work has improved upon PBT by using more advanced hyperparameter selection techniques (Parker-Holder et al., 2020). In contrast to SEARL, PBT and follow-ups do not optimize the architecture and, more importantly, do not share experience within the population. As our experiments show, these advances lead to speedups of up to 10x in terms of sample efficiency.

## 3 SAMPLE-EFFICIENT AUTORL

In this section, we introduce a *Sample-Efficient framework for Automated Reinforcement Learning (SEARL)* based on an evolutionary algorithm acting on hyperparameters and gradient-based training using a shared experience replay. First, we discuss relevant background that describes SEARL building blocks before giving an overview of the proposed AutoRL framework, followed by a detailed description of each individual component.

### 3.1 BACKGROUND

**Evolvable neural network:** Using evolutionary algorithms to design neural networks, called Neuroevolution (Floreano et al., 2008; Stanley et al., 2019), is a long-standing approach. Some approaches only optimize the network weights (Such et al., 2017), while others optimize architectures and weights jointly (Zhang & Mühlenbein, 1993; Stanley & Miikkulainen, 2002). To evolve the neural network in SEARL, we encode RL agent's neural architectures by the number of layers and the nodes per layer, similar to (Miikkulainen et al., 2019). When adding a new node to a layer, existing parameters are copied, and newly added parameters are initialized with a small magnitude. This is a common technique to preserve already trained network weights (Wei et al., 2016).

**Shared experience replay:** Replaying collected experiences (Lin, 1992; Mnih et al., 2015) smooths the training distribution over many past rollouts. The experience replay acts as a store for experience collected by agents interacting with the environment. Deep RL algorithms can sample from this storage to calculate gradient-based updates for the neural networks. It has been used and extended in various flavors, often to make use of diverse experience or experience collected in parallel (Horgan et al., 2018; Khadka & Tumer, 2018; Bodnar et al., 2020; Khadka et al., 2019). SEARL employs a shared experience replay, which stores the diverse trajectories of all differently configured RL agents in the population so that each individual can benefit from the collective experiences during training.

## 3.2 Framework

In SEARL, each individual in our population represents a deep reinforcement learning agent consisting of a policy and value network and the RL training algorithm hyperparameters, including the neural network architecture. The training and meta-optimization of these individuals take place in an evolutionary loop that consists of five basic phases (initialization, evaluation, selection, mutation, and training), as shown in Figure 1. During one epoch of this evolutionary loop, all individual properties could change through different mutations and training operators. This happens independently for each individual and can be processed in parallel.

A novel feature of our approach is that rollouts of each individual are not only used for evaluation and selection purposes but also serve as experience for off-policy training of all agents and are stored in a shared replay memory. When changing the architecture, we follow the approach of *Lamarckian* evolution (Ross, 1999), where the updated weights of an agent during training are not only used in the evaluation phase but are preserved for the next generation. In the following, we describe the different phases of our evolutionary loop in detail; we refer the reader to Appendix B for detailed pseudocode of the algorithm.

**Initialization:** SEARL uses a population $pop$, of $N$ individuals, each consisting of an RL agent $A_i$, and its hyperparameter settings $\boldsymbol{\theta}_i$. Thus, we can represent $pop_g$ at each generation $g$ as follows:

$$pop_g = (\{A_1, \boldsymbol{\theta}_1\}_g, \{A_2, \boldsymbol{\theta}_2\}_g, ..., \{A_N, \boldsymbol{\theta}_N\}_g) \tag{1}$$

The individual's hyperparameter setting $\boldsymbol{\theta}_i$ is composed of architecture hyperparameters, such as the number of layers or the layer sizes, and algorithm hyperparameters, such as the learning rate. To arrive at a minimum viable neural network size, we start with reasonably small neural network architecture and enable its growth by using mutation operators. Other hyperparameters are set to some initial value, as they are subsequently adapted in the evolutionary loop. In our experiments we observed that using random initialization of hyperparameters for SEARL did not lead to large performance differences, see Appendix E. This suggests that there is very little domain-specific knowledge required to effectively use SEARL.

**Evaluation:** After initialization and after each training phase, we evaluate each individual in the population using the RL agent, $A_i$, for at least one episode or a minimum number of steps in the environment. This ensures a minimum amount of new experience from each agent and keeps the stored trajectories in the shared replay memory diverse. The evaluation can be performed in parallel since each agent acts independently. We use the mean reward of the individual's evaluation as fitness value $f_i$.

**Selection:** We use tournament selection with elitism (Miller & Goldberg, 1995) for each prospective slot in the population of the new generation. For each tournament, $k$ individuals are randomly chosen from the current population $pop_g$, and the individual with the largest fitness value $f_i$ is selected for the slot. We repeat this tournament $N-1$ times to fill all slots. The size $k$ allows us to control how greedily the selection mechanism picks candidates for the new population. We reserve one spot in the new population for the current population's best-performing individual, thus preserving it across generations.

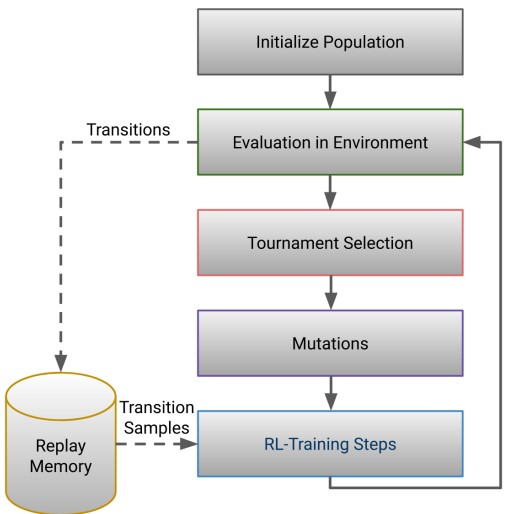

Figure 1: Overview of Sample-efficient AutoRL (SEARL).

**Mutation:** To explore the space of network weights and hyperparameters, we use different single-parent mutation operators. We apply one of the following operators uniformly at random to each member: (1) Mutation of the weights of the neural networks by adding Gaussian noise. (2) Change of activation function of the neural network. (3) Change of the neural network size by either adding

additional nodes to a given layer or adding a new layer altogether while reusing the trained weights and initializing new weights randomly. (4) Change of algorithm hyperparameters. (5) No operation. We refer the reader to Appendix A for more details.

**Training:** Using each individual's current hyperparameters, we train it by sampling from the shared replay memory. Each individual is trained for as many steps as frames have been generated in the evaluation phase, as is common practice in deep RL. Optionally, the training time could be reduced by using only a fraction $j$ of the steps to adapt to computational constraints. Since the neural network size could be subject to mutation between two training phases, the target network of the RL algorithm needs to be adapted too. Furthermore, the optimizer weight-parameters connected to individual network weights cannot remain the same across generations. We address these issues by creating a new target network and re-initializing the optimizer at the beginning of each training phase. Our experiments show that this re-creation and re-initialization does not harm the performance of the considered RL algorithm. Please find more details in Appendix C. Like the evaluation phase, the training can be performed in parallel since every individual is trained independently.

**Shared experience replay:** A shared experience replay memory collects trajectories from all evaluations and provides a diverse set of samples for each agent during the training phase. This helps to improve training speed and reduces the potential of over-fitting.

## 4 CASE STUDY: META-OPTIMIZATION OF TD3

In this section, we demonstrate the capabilities and benefits of SEARL. We simulate the meta-optimization of the Twin Delayed Deep Deterministic Policy Gradient algorithm (TD3; Fujimoto et al., 2018) on the widely used MuJoCo continuous control benchmark suite (Todorov et al., 2012). A well-performing hyperparameter configuration on MuJoCo is known, even though the optimization of TD3's hyperparameters is unreported. This makes up a useful setting to investigate meta-optimization with SEARL compared to unknown algorithms or environments.

To study SEARL, we assume *zero* prior knowledge about the optimal hyperparameter configuration and compare SEARL against two plausible baselines. All meta-algorithms are tasked with optimizing network architecture, activations, and learning rates of the actor and the critic. Following Fujimoto et al. (2018), we evaluate all methods on the following MuJoCo environments: `HalfCheetah`, `Walker2D`, `Ant`, `Hopper` and `Reacher`. In general, other off-policy RL algorithms could also be optimized using SEARL due to the generality of our framework.

### 4.1 BASELINES

Since there is no directly comparable approach for efficient AutoRL (see Section 2), we compare SEARL to random search and a modified version of PBT that allows for architecture adaptation. As RL is sensitive to the choice of hyperparameters, the design of the configuration space matters. To allow for a fair comparison, we keep the search spaces for all methods similar; see Appendix D.

**Random Search:** Random search makes no assumptions on the algorithm and is a widely used baseline for HPO (Feurer & Hutter, 2019). We sample 20 random configurations and train TD3 using each configuration without any online architecture or hyperparameters changes for $1M$ frames. We select the configuration that resulted in the best training performance and evaluate it with 10 different random seeds to obtain its validation performance.

**Modified Population-Based Training:** To fairly compare PBT with our proposed approach, we modify PBT to be able to adapt the neural architecture of the networks. We perform the same mutations to the activation function and neural network size as in SEARL (Section 3.2). This modification is required for a fair comparison since using an arbitrary fixed network size could compromise PBT's performance. On the other hand, using the network size reported by Fujimoto et al. (2018) would not take into account any potential HPO due to an unknown budget. In other aspects, we follow Jaderberg et al. (2017), e.g., initializing the population by sampling from the configuration space uniformly at random. We train and evaluate a population of 20 agents for 100 generations. In every generation, the agents are trained and evaluated for 10 000 steps each.

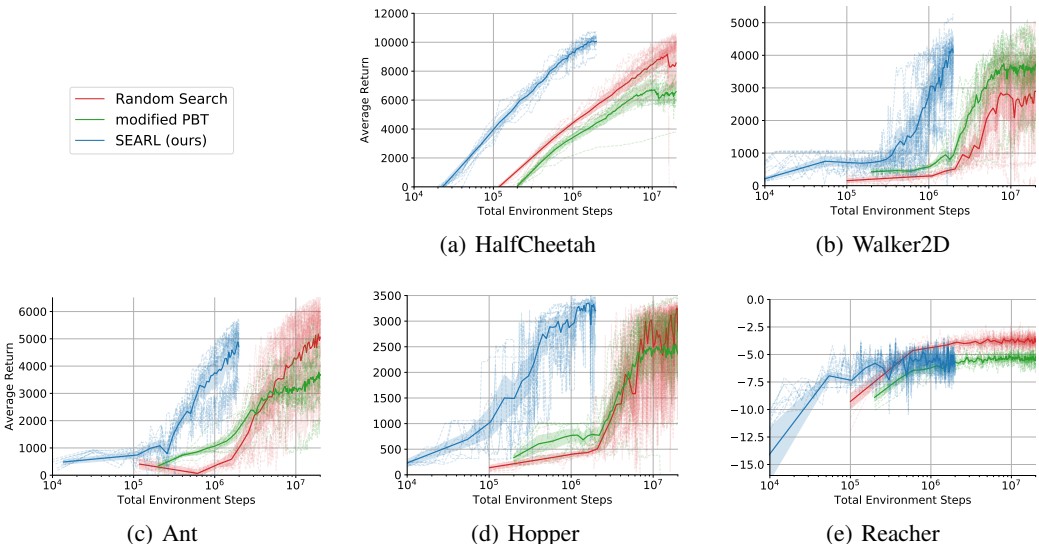

(a) HalfCheetah

(b) Walker2D

(c) Ant

(d) Hopper

(e) Reacher

Figure 2: Performance comparison of SEARL, modified PBT, and random search regarding the required total steps in the environment. Random search and modified PBT require an order of magnitude more steps in the environment to reach the same reward as SEARL. The blue line depicts SEARL, the green line of modified PBT, and the red line of random search. All approaches are evaluated over 10 random seeds. The x-axis shows the required environment interactions in log-scale. Dashed lines are performances of the different seeds and the shaded area of the standard deviation.

## 4.2 SEARL FOR TD3

Since we evolve actor and critic networks over time, we start with a one-layer neural network with 128 hidden nodes, which represents the lower bound of the search space defined in Appendix D. In an experiment investigating random initialization, we found no benefit of a more diverse starting population; please find details in Appendix E. For the remaining hyperparameters, we chose a commonly used configuration consisting of $ReLU$ activation. We use the Adam default value ($1e^{-3}$) (Kingma & Ba, 2015) as an initial learning rate. Similar to the PBT baseline, we used 20 workers.

We evaluate using one complete episode for rollouts or at least 250 steps in the environment. This results in at least 5 000 new environment interactions stored in total in the shared replay memory each generation. To select a new population, we use a tournament size of $k = 3$. After mutation, we train the individuals with samples from the shared replay memory for $j = 0.5$ times the number of environment interactions of the whole population during evaluation in this generation. We run SEARL for a total of 2 million environment interactions. This results in 1 million training updates for each individual in the population, similar to random search and PBT. We provide a detailed table of the full configuration in Appendix F.

## 4.3 FAIR EVALUATION PROTOCOL FOR HPO IN RL

Standard evaluation of RL agents commonly shows the achieved cumulative reward in an episode after interacting with the environment for $N$ steps. It is common practice to only report the best configuration's rewards found by an offline or online meta-optimization algorithm. However, this hides the much larger number of environment interactions needed to find this well-performing configuration. In contrast, we argue that the interaction counter should start ticking immediately as we tackle a new task or environment. Such a fair reporting of meta-optimization could also increase the comparability of approaches (Feurer & Hutter, 2019).

As there is no convention on how to display the performance of meta-learning while taking into account the total amount of environment interactions as described above, we opt for the following simple presentation: When reporting the meta-optimization effort, every environment interaction

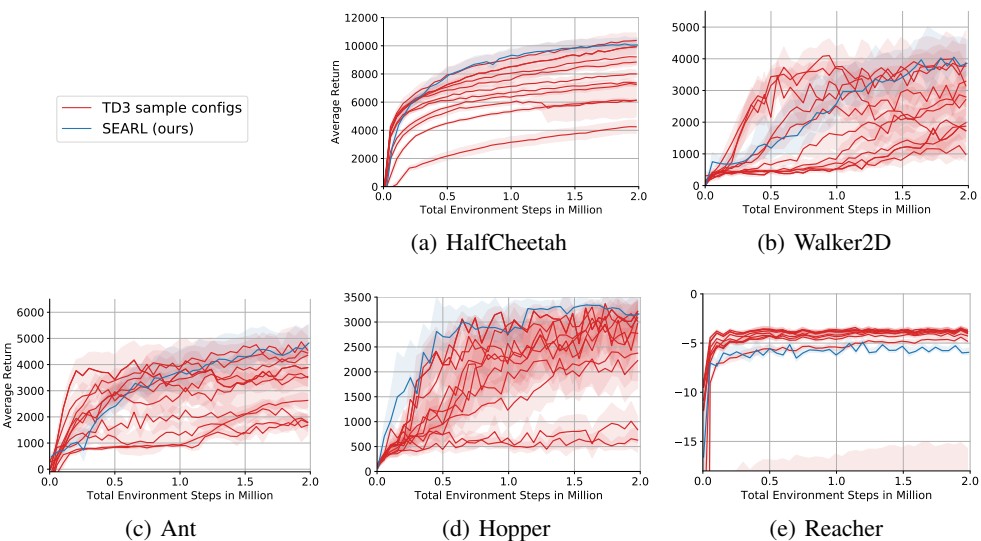

Figure 3: Comparison of the mean performance over 10 random seeds of SEARL (blue line) against 10 randomly sampled configurations of TD3, evaluated over 10 random seeds (red) over 2M environment frames. The shaded area represents the standard deviation.

used for the meta-optimization process should be counted when reporting the performance at any given time. In the case of offline meta-optimization efforts as in random search trials, this translates to multiplying the number of interactions of the best configuration's performance curve by the number of trials used to arrive at this configuration.

## 5 RESULTS & DISCUSSION

**Comparison to baselines:** In Figure 2, we compare the mean performance of SEARL to random search and our modified version of PBT when training the TD3 agent and tuning its algorithm and architecture hyperparameters. We report the results following the evaluation outlined in 4.3.

SEARL outperforms random search and modified PBT in terms of environment interactions with up to $10\times$ faster convergence in each task, which indicates a significant gain in the sample efficiency of our approach. We want to stress that individuals within the population do not get to use single experience samples more often during training compared with the modified PBT setup. The substantial increase in sample efficiency for meta-optimization is mainly due to the fact that individual experience samples get re-used more often during the training phase across the whole population. This gained efficiency of the individual learning agents can compensate the overhead of the meta-optimization, which allows us to get a strong hyperparameter configuration in only two times the environment interactions compared to the reported *final* performance in the TD3 paper.

Furthermore, in Figure 2, we see that SEARL outperforms random search in two and modified PBT in four out of five tasks in terms of the final performance, while clearly dominating in terms of anytime performance. In the other tasks, SEARL is on par with the baselines, and we often see a smaller standard deviation in performance, which could indicate that SEARL is robust to random seeds. However, SEARL and PBT struggles on `Reacher` in comparison to random search. The lower performance on `Reacher` could be attributed to the very short episode length of this environment. TD3 can cope with this setup due to frequent training updates. However, in SEARL and other evolutionary approaches like ERL, where a fixed number of samples is collected, this could harm the performance since the training updates become less frequent.

**Performance in the Search Space:** In Figure 3, we evaluate the single-run performance of SEARL in the search space of TD3 random search. We plot the mean performance over 10 random seeds for a single SEARL run and 10 sampled TD3 configurations over 10 random seeds each. The TD3 configurations run for $2M$ environment frames similar to SEARL and in contrast to $1M$ frames in

the upper experiment. We observe that SEARL matches the performance of the favorable part of the search space even within the same number of environment steps as a single TD3 run, again only struggling on `Reacher`. This indicates that SEARL evolves effective configuration schedules during training, even when starting with a population consisting of a single hyperparameter setting using a small one-layer neural network architecture.

**Dynamic Configuration:** We show exemplary hyperparameter schedules discovered by SEARL on the HalfCheetah task in Figure 4. In the first $500\,000$ environment interactions, SEARL quite quickly increases the network size of the TD3 agents to values, which are more in line with commonly reported network architectures used for strong results on MuJoCo's HalfCheetah task. Afterward, the architecture size is increased more slowly, before saturating after $\sim 1$M environment interactions. This shows SEARL can adapt the network architecture starting from a relatively small network size and prevent further increases in size once the network capacity is sufficient. The learning rates are shown in the lower part of Figure 4 for both the actor and the critic networks, and can be seen to slowly decrease over time. Different schedules can be observed on other environments, see Appendix G. The need for individual adaptation in each environment is also reflected by the final neural network architectures of the best-performing agent, as shown in Appendix I.

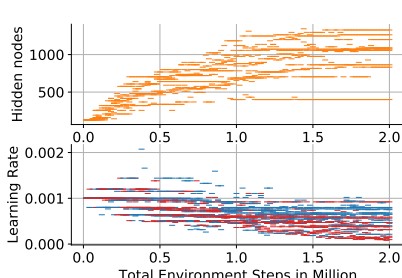

Figure 4: SEARL changes of network size (top) and actor/critic learning rate (bottom, red/blue) across the population in the HalfCheetah environment.

While the final actor networks in `HalfCheetah` have on average 2.8 layers and more than $1\,000$ nodes, for other environments such as `Walker2d` or `Ant` smaller network sizes perform better. This suggests that SEARL could be capable of automatically adapting the network size to the task complexity. Another advantage of the dynamic configuration in SEARL lies in the possibility to achieve on-par performance reported in Fujimoto et al. (2018)[TD3] with significantly smaller network architectures. Experiments with smaller static network architectures in TD3 do not achieve comparable performance. This suggests that growing networks like in SEARL is helpful for achieving strong performance with small network sizes.

**Ablation Study:** To determine how the individual parts of our proposed method influence performance, we conducted an ablation study, which is shown in Figure 5. For the ablation experiment without architecture evolution we set the hidden layer size to $[400, 300]$. The ablation study clearly shows that disabling shared experience replay harms the performance most severely. Also, the other aspects of SEARL are beneficial in the mean. The various environments benefit from the individual SEARL features to different extents. While some tasks benefit strongly from architecture adaptation, others benefit more from hyperparameter changes. For example, removing the architecture adaptation in HalfCheetah leads to a drop in performance while it is beneficial in Walker2D, while the hyper-parameter adaptations exert the opposite behavior.

To test the generality of SEARL, we conducted a held-out experiment. SEARL performs better or on-par on four out of five environments compared with a held-out tuned TD3 configuration, demonstrating the generalization capabilities of SEARL across different environments. Details can be found in Appendix H.

**Meta-optimization of DQN:** To demonstrate the generalization capabilities of our proposed framework, we also use SEARL to automatically tune the widely used Rainbow DQN algorithm (Hessel et al., 2018) on five different environments of the Arcade Learning Environment (Bellemare et al., 2013) and assume zero prior knowledge about hyperparameters. Following the approach outlined in Section 3, we initialized the neural networks starting from a small size and allowed changes in the CNN & MLP activation functions as well as the learning rate during the mutation phase. Please find a detailed description in Appendix J. We compare a SEARL-optimized and a RS-optimized Rainbow DQN and observe similar results as in the case study on TD3, where SEARL shows a significantly increased sample-efficiency, as well as a strong final performance, surpassing the best random search configuration's performance in all but one environment (see Figure 10, Appendix J). This suggests that SEARL is capable of optimizing off-policy RL algorithms across various environments with different input modalities, as well as different RL algorithm setups.

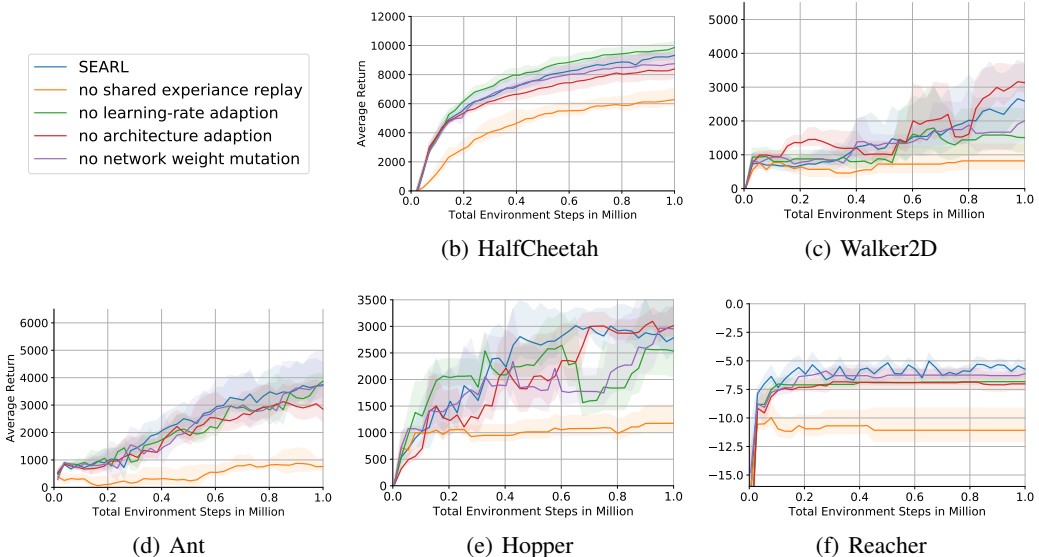

Figure 5: Comparison of the performance impact when removing different SEARL features individually. All results show mean performance across five different random seeds. The standard deviation is shown as shaded areas.

**SEARL for *On-Policy* Deep RL:** The meta-optimization of on-policy algorithms using SEARL would most likely *not* show the same benefits in sample efficiency due to the high impact of the shared experience replay. Nonetheless, we expect that the dynamic configuration of the algorithm and network architecture hyperparameters would still positively impact on-policy algorithms. Another option would be to incorporate the choice of the deep RL algorithm being optimized as an additional hyperparameter for each individual, thus allowing multiple RL algorithms to be optimized.

**Computational Effort:** SEARL requires less compute to optimize an agent than a random search or PBT. This has two reasons: (A) As shown in Figure 2, SEARL requires up to 10 times fewer environment interactions. (B) SEARL evolves a neural network starting with a small network size, which reduces computational costs at the beginning of the meta-optimization. Even though all compared methods can run in parallel, random search and PBT can run asynchronously, while SEARL has an evolutionary loop that requires synchronicity, at least in our current implementation.

## 6 CONCLUSION

Advances in AutoRL dispose of the required manual tuning of an RL algorithm's hyperparameters. Besides the general benefits of automated optimization, like fair comparability of different algorithms (Feurer & Hutter, 2019), more automation in the HPO of RL algorithms leads to improved sample efficiency. This could make algorithms more viable in real-world applications.

Following these desiderata, we proposed SEARL, a sample-efficient AutoRL framework. SEARL performs an efficient and robust training of any off-policy RL algorithm while simultaneously tuning the algorithm and architecture hyperparameters. This allows SEARL to discover effective configuration schedules, addressing the non-stationarity of the RL problem. We demonstrated the effectiveness of our framework by meta-optimizing TD3 in the MuJoCo benchmark suite, *requiring up to an order of magnitude fewer interactions* in the environment than random search and PBT while matching or improving their final performance in 4 out of 5 environments.

The improvement in sample efficiency can be attributed mainly to the use of shared replay experiences when training the population of agents and allows us to perform AutoRL almost *for free*. SEARL required only twice the number of environment interactions as a single TD3 run (without HPO) while simultaneously training agents and performing a dynamic hyperparameter optimization.

## ACKNOWLEDGMENT

The authors acknowledge funding by the Robert Bosch GmbH, as well as support by the state of Baden-Württemberg through bwHPC and the German Research Foundation (DFG) through grant no INST 39/963-1 FUGG. This work has also been supported by the European Research Council (ERC) under the European Union's Horizon 2020 research and innovation programme under grant no. 716721.

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

APPENDIX

## A    MUTATION OPERATORS

For each individual, we select one of the following operators uniformly at random:

**Gaussian Mutation:**  This operator adds Gaussian noise to a subset of the RL agent's neural network weights. The additive noise is only added to the actor network, in a similar fashion, as described in Khadka & Tumer (2018) Algorithm 3, using the same hyperparameters.

**Activation Function:**  For each network independently, a new activation function for the hidden layers is chosen randomly from a pre-defined set with equal probability. The set contains the activation functions (**relu**, **elu**, **tanh**) excluding the current used activation function.

**Network Size:**    This operator changes the network size, either by adding a new layer with a probability of 20% or new nodes with a probability of 80%. When adding a new layer, the last hidden layer will be copied, and its weights are re-initialized. This leaves the features learned in the earlier layers unchanged by the mutation operator. When adding new nodes, we first select one of the existing hidden layers at random and add either 16, 32 or, 64 additional nodes to the layer. The existing weight matrix stays the same, and the additional part of the weight matrix is initialized.

**Hyperparameter Perturbations:**    This operator adapts the hyperparameters of the training algorithm online. We make use of hyperparameter changes as proposed in PBT Jaderberg et al. (2017). For the TD3 case study, we only change the learning rates of the actor and critic network. To this end, we randomly decide to increase or decrease the learning rate by multiplying the current rate by a factor of 0.8 or 1.2.

**No-Operator:**    No change is applied to the individual. This option allows more training updates with existing settings.

# B SEARL ALGORITHM

---

**Algorithm 1:** SEARL algorithm

---

**Input:** population size $N$, number of generations $G$, training fraction $j$

**1** Initialize replay memory $\Re$

**2** Initialize start population $pop_0 = (\{A_1, \boldsymbol{\theta}_1\}_0, \{A_2, \boldsymbol{\theta}_2\}_0, ..., \{A_N, \boldsymbol{\theta}_N\}_0)$

**3** **for** $g = 1$ **to** $G$ **do**

**4**    Set fitness list $\mathcal{F} = ()$

**5**    Set selected population $\widetilde{pop}^{selected} = ()$

**6**    Set mutated population $\widetilde{pop}^{mutated} = ()$

**7**    Set transition count $\tau = 0$

    /* Evaluation of previous population                            */

**8**    **foreach** $\{A_i, \boldsymbol{\theta}_i\}_{g-1} \in pop_{g-1}$ **do**

**9**      $f_g^i$, transitions $\leftarrow$ EVALUATE($\{A_i\}_{g-1}$)

**10**      store transitions in $\Re$

**11**      $\tau \mathrel{+}= len(\text{transitions})$

**12**      add $f_g^i$ to $\mathcal{F}_g$

**13**    **end**

    /* Tournament selection with elitism                         */

**14**    $best \leftarrow$ SELECTBESTINDIVIDUAL($pop_{g-1}, \mathcal{F}_g$)

**15**    add $best$ to $\widetilde{pop}^{selected}$

**16**    **for** $i = 1$ **to** $N - 1$ **do**

**17**      $winner_i \leftarrow$ TOURNAMENTSELECT($pop_{g-1}, \mathcal{F}_g$)

**18**      add $winner_i$ to $\widetilde{pop}^{selected}$

**19**    **end**

    /* Mutation of tournament selection                          */

**20**    **foreach** $\{A_i, \boldsymbol{\theta}_i\} \in \widetilde{pop}^{selected}$ **do**

**21**      $offspring_i \leftarrow$ MUTATE($\{A_i, \boldsymbol{\theta}_i\}$)

**22**      add $offspring_i$ to $\widetilde{pop}^{mutated}$

**23**    **end**

    /* RL-Training steps                                         */

**24**    **foreach** $\{A_i, \boldsymbol{\theta}_i\} \in \widetilde{pop}^{mutated}$ **do**

**25**      sample $transitions$ from $\Re$

**26**      $trained_i \leftarrow$ RL-TRAINING($\{A_i, \boldsymbol{\theta}_i\}, transitions$, training_steps$= \tau * j$)

**27**      add $trained_i$ to $pop_g$

**28**    **end**

**29** **end**

---

## C   IMPACT OF TARGET NETWORK RE-CREATION AND OPTIMIZER RE-INITIALIZATION

Due to the changing network architecture in the mutation phase of SEARL, each training phase requires a change of the value target network and a re-initialization of the optimizers used during training. To deal with this, SEARL copies the individual's current value network after the mutation phase and uses this copy as the frozen value target network for the training phase. SEARL also re-initializes the optimizers to deal with the changing network architectures. We analyze the performance impact of this *target network re-creation and re-initialization* by simulating the effects in the context of the TD3 algorithm in the MuJoCo suite. To this end, we re-create the target network and re-initialize the optimizer after every episode in the environment. We note that in the original TD3 setting, the target network is updated using soft updates. The target re-creation case we analyze corresponds to a hard update of the target network. The results of the described experiments are shown in Figure 6, where each curve represents the mean performance of 20 TD3 runs with different random seeds, and the shaded area represents the standard deviation. We observe that, perhaps surprisingly, re-creation and re-initialization do not seem to hurt performance and even improve performance in 2 out of 5 environments.

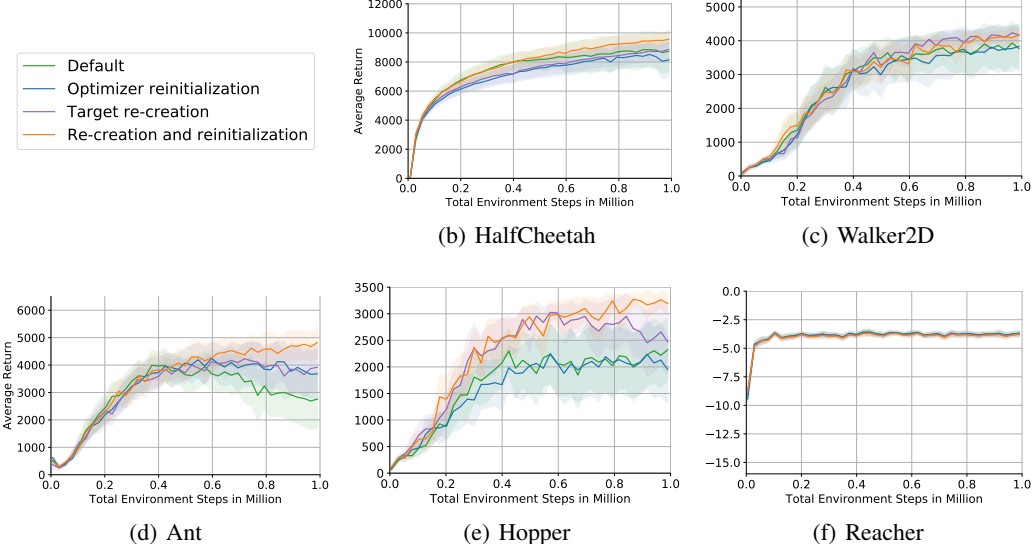

Figure 6: This plot shows the impact of target re-creation and optimizer re-initialization after each episode in the TD3 algorithm. The default configuration uses a soft target update and only initializes the optimizer once at the beginning of the training. The plots show the mean performance across 20 random seeds on the MuJoCo suite, and the shaded area represents the standard deviation.

# D  HYPERPARAMETER SEARCH SPACES

The hyperparameter search space is inspired by the PBT-UNREAL experiments and can be found in Table 1. We choose the initial hyperparameter space for PBT smaller than random search since PBT can increase or decrease parameters during the optimization.

| Parameter | PBT | Random Search |
|---|---|---|
| learning rate | $\{5 \times 10^{-3}, 10^{-5}\}$ | |
| activation | $\{\mathrm{relu}, \mathrm{tanh}, \mathrm{elu}\}$ | |
| layers | $\{1, 2\}$ | $\{1, 2, 3\}$ |
| units | $(128, 384)$ | $(64, 500)$ |

Table 1: Considered hyperparameter search spaces. The random search space is slightly larger since the configurations are fixed, whereas PBT can explore network sizes.

# E    RANDOM HYPERPARAMETER INITIALIZATION IN SEARL

One may hypothesize that a more diverse starting population coming from random initialization of each individual might help to find well-performing configurations faster. In this experiment we initialize the SEARL population with the same search space as in PBT (Appendix D). Figure 7 shows the mean performance of 10 runs with different random seeds. The shaded area represents the standard deviation. We observe that using a more diverse starting population does not provide substantial benefits in any environment. But in cases where less task-specific information is available, we show that using a random initialization does not seem to harm performance significantly.

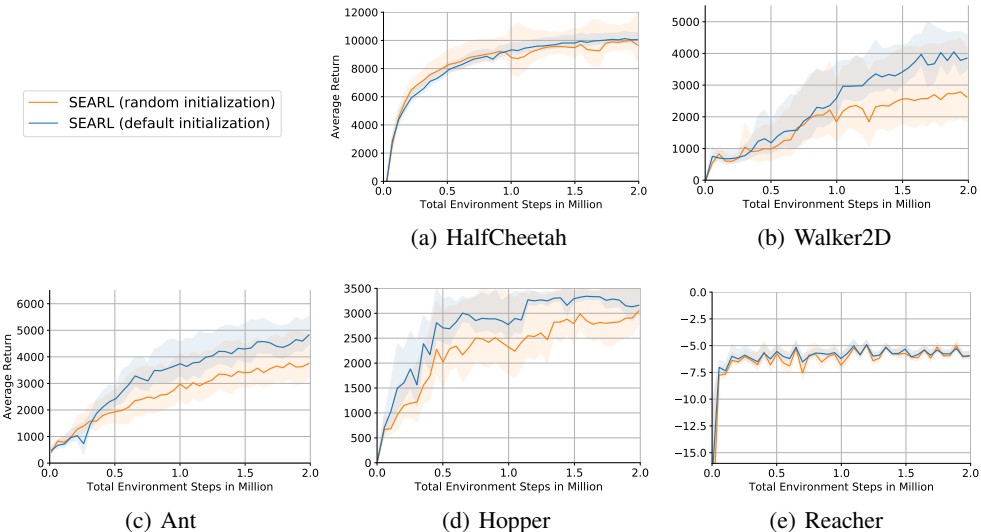

Figure 7: The mean performance over 10 random seeds of SEARL using the default configuration for hyperparameter initialization (blue line) compared with random initialization (orange) over 2M environment steps. The shaded area represents the standard deviation.

## F  SEARL CONFIGURATION

Table 2 shows the configuration of SEARL for the TD3 case study.

| Parameter | Value |
|---|---|
| Max. frames in environment | 2 000 000 |
| Replay memory size | 1 000 000 |
| Min. frames per evaluation | 250 |
| Population size | 20 |
| Selection tournament size | 3 |
| New layer probability | 20% |
| New nodes probability | 80% |
| Parameter noise standard deviation | 0.1 |
| Training batch size | 100 |
| Fraction of eval frames for training | 0.5 |
| Default learning rate | 0.001 |
| Optimizer | Adam |
| TD3 Gamma | 0.99 |
| TD3 Tau | 0.005 |
| TD3 policy noise | 0.2 |
| TD3 noise clip | 0.5 |
| TD3 update frequency | 2 |
| Default activation function | **relu** |
| Start network size | [128] |

Table 2: The configuration of SEARL in the TD3 case study.

# G  CHANGE OF HYPERPARAMETERS OVER TIME

We visualize a set of hyperparameters for a SEARL optimization run to generate more insights about the online hyperparameter adaptation. We show the sum of hidden nodes and the learning rates of both the actor and the critic for all individuals in the population in Figure 8. While for runs in environments like HalfCheetah and Ant, the learning rate seems to decrease over time for most individuals. In other environments, the learning rate seems to stay more constant. This suggests that no single learning rate schedule seems optimal for all environments, thus showing the need for online hyperparameter adaptation. We also observe that different environments require different network sizes to achieve strong performance. This is in line with previous findings, e.g., in Henderson et al. (2018).

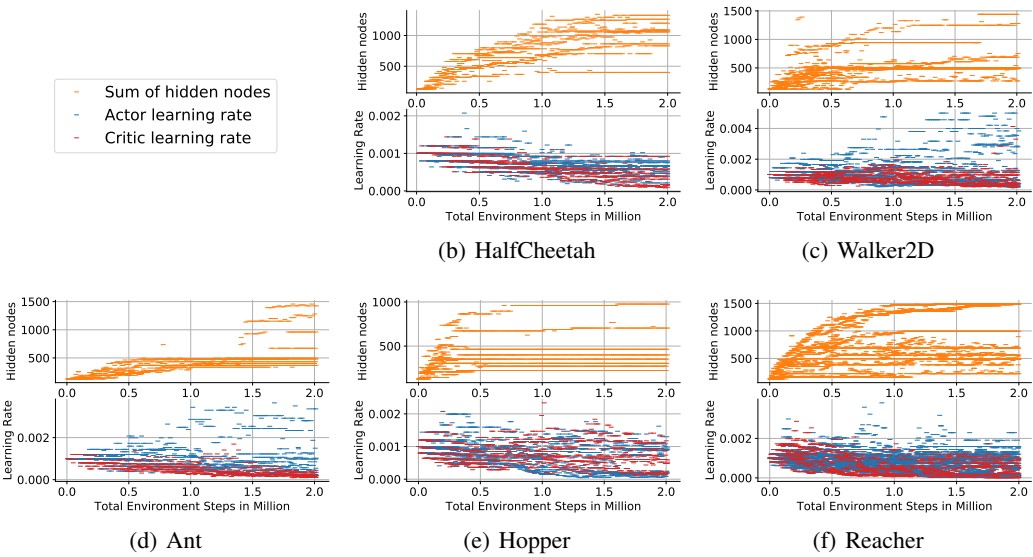

Figure 8: Change of the total network size (as the sum of hidden nodes in all layers, orange) and the learning rates (actor learning rate in red, critic learning rate in blue) over time. A plot contains all population individuals configurations over 2M steps for one SEARL run.

# H   SEARL COMPARED TO TD3 HELD-OUT OPTIMIZATION

We test the generality of SEARL against a held-out tuned TD3 configuration. Therefore, we tune TD3 hyperparameters in the artificial setting of four similar environments with 2M interactions in total to find the hyperparameter setting that is best on average across these four environments and evaluate on the held-out environment for 2M steps. We compare this leave-one-environment-out performance to a single fixed hyperparameter configuration for SEARL across all environments running for 2M environment interactions to demonstrate that SEARL is capable of robust joint HPO and training of a learning agent. As shown in Figure 9, SEARL outperforms the TD3-tuned hyperparameter setting in three environments, performs on par in "Walker2d" and slightly worse in "Reacher". These results demonstrate the generalization capabilities of SEARL across different environments.

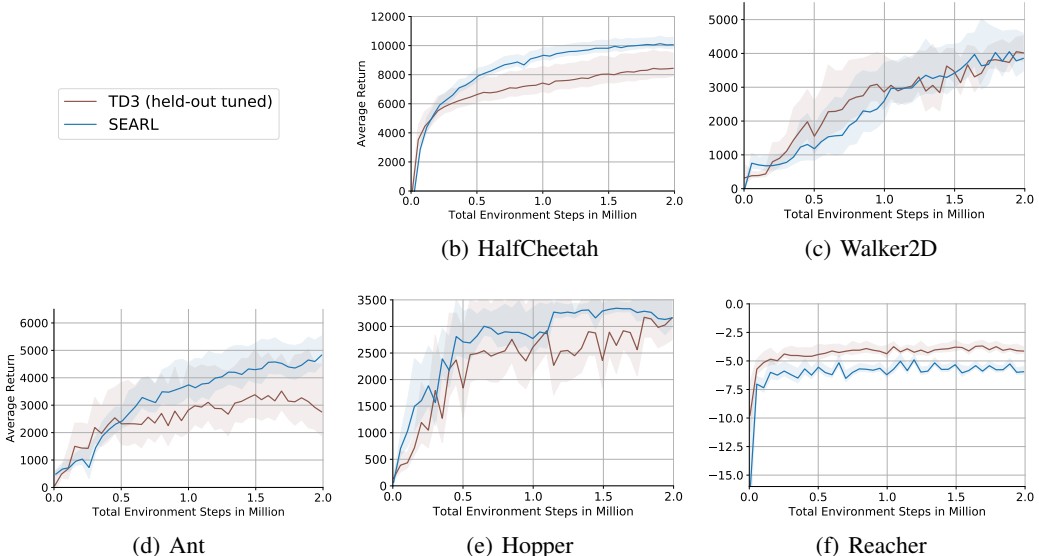

Figure 9: The comparison of SEARL performance against a held-out tuned TD3 configuration, tuned with the same budget of a SEARL run (2M environment interactions), and evaluated over 2M steps. All results show mean performance across 10 different random seeds. The standard deviation is shown as shaded areas.

# I   FINAL ARCHITECTURE FOUND BY SEARL IN TD3 MUJOCO EXPERIMENTS

Table 3 shows the average across ten different seeded runs of the architecture of the best performing individual. While the actor networks in `HalfCheetah` have 2.8 layers and over $1\,000$ nodes on average the networks are much smaller in other environments like `Walker2d` or `Ant`. This could indicate that different environments require different network capacities. SEARL could be capable of automatically adapting to the network size to the task complexity.

| MuJoCo | Avg. Layer | Avg. total nodes | Avg. nodes per layer |
|---|---|---|---|
| HalfCheetah | 2.8 (0.1) | 1019.2 (80.3) | 360.8 (22.3) |
| Walker2d | 1.7 (0.2) | 689.6 (113.3) | 422.7 (34.4) |
| Ant | 1.7 (0.2) | 723.6 (119.5) | 432.59 (16.6) |
| Hopper | 1.3 (0.1) | 462.0 (75.5) | 357.0 (27.0) |
| Reacher | 2.1 (0.3) | 840.0 (145.9) | 394.1 (38.7) |

Table 3: Comparing the average final architecture across 10 random seeds in the different MuJoCo environments. The standard error in brackets.

## J    OPTIMIZING RAINBOW DQN WITH SEARL ON ATARI

To demonstrate the generalization capabilities of the proposed SEARL framework to different RL settings, we perform AutoRL tuning of the widely used Rainbow DQN algorithm (Hessel et al., 2018) in four Atari environments (Bellemare et al., 2013). Rainbow DQN is a combination of different extensions to the DQN algorithm (Mnih et al., 2015). We use all the extensions proposed in Rainbow DQN except for the prioritized replay memory (Schaul et al., 2016) due to the shared experience collection in SEARL. We compare SEARL to random search to evaluate SEARL's performance in the search space of this setup. We report the results in Figure 10.

For the random search experiments, we select the best performance out of 10 sampled configurations. For each configuration, we randomly choose the CNN layer count as well as the channel size, stride and kernel size, the MLP layer count and size, the activation function, and learning rate, please find details in Table 4. Compared to the TD3 case study in section 4, the SEARL Rainbow DQN setup differs only in the agents' initial network size. Here we use a small MLP $[128]$ and a small CNN (two-layer, channel size $[32, 32]$, kernel size: $[8, 4]$, stride: $[4, 2]$). To evolve the agents, we allow changes to the MLP by adding layer or nodes to existing layer (add $\{8, 16, 32\}$), activation function ($\{relu, tanh, elu\}$), and the learning rate during the mutation phase, similarly to the TD3 case study. In addition, we mutate the CNN by adding new convolution layers or changing the channel size (add $\{8, 16, 32\}$), kernel size ($\{3, 4, 5\}$) or stride ( $\{1, 2, 4\}$ and in sum 8).

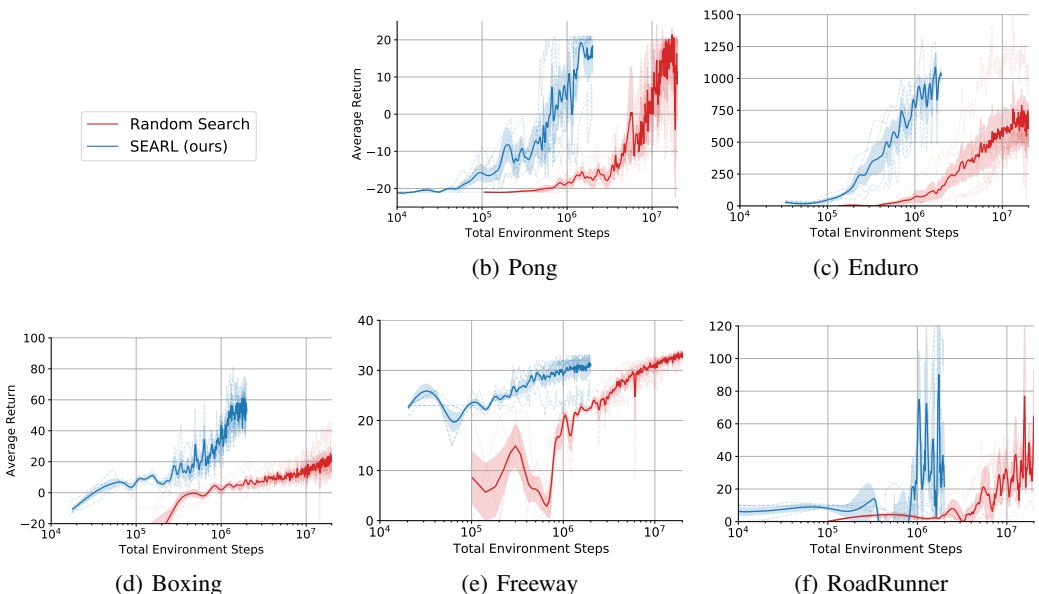

Figure 10: Performance comparison of SEARL and random search regarding the required total steps in the environment. Random search requires an order of magnitude more steps in the environment to reach the same reward level as SEARL. The blue line depicts SEARL and the red line random search. SEARL is evaluated over 5 random seeds and random search over 5 runs with 10 configurations each. The x-axis shows the required environment interactions in log-scale. Dashed lines are performances of the different seeds and the shaded areas show the standard deviation.

We run both SEARL and each random search configuration for 2M frames in the environment. We report results in the same fashion as for the TD3 case study. For each environment, we perform 5 SEARL runs with different random seeds and 5 runs of random search (which choose the best configuration out of 10 random configurations each). We report the best of these runs scaled by the total required step cost. As previously observed in the TD3 case study, we observe that SEARL again outperforms the random search baseline significantly in terms of sample-efficiency. In Pong and RoadRunner, the SEARL optimized performance is on par with the best random search configuration, and in Freeway, the SEARL optimized final performance is slightly worse. Still, in both Enduro and Boxing, SEARL outperforms the best random search configurations.

These results demonstrate the general usefulness of SEARL across different RL algorithms and environments, even covering different types of network architectures and input modalities.

| Parameter | Random Search |
|---|---|
| learning rate | $\left\{5 \times 10^{-3}, 10^{-5}\right\}$ |
| activation | $\{relu, tanh, elu\}$ |
| MLP layers | $\{1, 2, 3\}$ |
| MLP nodes | $[64, 384]$ |
| CNN layers | $\{1, 2, 3\}$ |
| CNN channels | $[64, 384]$ |
| CNN stride | $\{1, 2, 4\}$ and $\sum_i^{layer} stride_i = 8$ |
| CNN kernel | $\{3, 4, 5\}$ |

Table 4: Considered hyperparameter search spaces for DQN experiments.

