# OpenReview forum: "Sample-Efficient Automated Deep Reinforcement Learning"
_ICLR.cc/2021/Conference — ICLR 2021 Poster_

### Official Review · AnonReviewer2 · 2020-10-28
**Experiments are not convincing**

**Rating:** 5
**Confidence:** 3

**Review:**

# Summary

The authors in this paper propose to optimize the hyperparameters and also the neural architecture while simultaneously training the agent. They evaluate the proposed method with TD3 in the MuJoCo benchmark suite.

Overall, the proposed method is well-motivated and well-written, and they provide enough experiment/implementation details to reproduce the results. More importantly, they provide the source code.

# Strength

- Tuning architecture and hyperparameters with PTB for off-policy RL has not been studied before. Compared to baselines, the proposed method is much more sample-efficient.
- The evaluation on MuJoCo is comprehensive, especially the ablation study in Section 4.

# Weakness

- They only test on a single benchmark with one method, TD3. MuJoCo is arguably simple. Actually, the visual world in MuJoCo is quite limited, so the encoder of the RL agent does not have to be huge. It could be more convincing if the authors can test on another benchmark, e.g. ProcGen.
- I think compared to computer vision tasks with huge neural networks, the search space for the architecture of RL models is much smaller, which can be observed in the ablation study. Without architecture adaption, there is no observable difference. It seems that combining the tuning of architecture and hyperparameters is not that useful.

---

> ### Author Response · Authors · 2020-11-18
> **Response to AnonReviewer2**
>
> We thank the reviewer for mentioning the novelty of our approach and the comprehensive evaluation.
>
> We fully understand the wish for more experiments on different off-policy algorithms and environments. However, ProcGen seems too expensive for a detailed evaluation in an AutoRL setting since we train at least 10 agents in parallel (especially for the short time window allocated for rebutting). As an alternative, also in line with requests by other reviewers, we implemented the popular Rainbow-DQN [1] on Atari environments, meta-optimized with the SEARL algorithm. These experiments are expensive as well. Thus, we do not expect to be able to finish the full 200million frames experiments commonly reported for Atari [1] during the short author response period. But we do believe this would be a valuable experiment and will include it in the final version of the paper.
>
> Regarding the impact of architecture adaption:
> Architecture choices have been shown to be important in RL. For example, Henderson et al. [2] experimentally evaluated the impact of the network architecture choices and found that it does impact the performance of the trained agents. Further, our own ablation study indicates that architecture adaptation is sometimes very beneficial. Specifically, without architecture adaptation, there is a significant performance drop in HalfCheetah and Ant. We updated the paper and included our ablation study from Appendix H into the main paper.
> We also added a paragraph in the main body, plus a new Appendix I, in which we focus on the found architectures. SEARL finds, e.g., for HalfCheetah larger actor-network sizes ( in average 2.8 layers/1000 nodes per network) compared to Walker2d ( in average of 1.7layers/680nodes per network). This could indicate that different environments require different network capacities. We also note that experiments with smaller, static network architectures in TD3 don't achieve the same performance as the setting reported in TD3. This suggests that growing networks like in SEARL constitutes achieving strong performance with small network sizes.
>
> Again, thank you a lot for your review!
>
> [1] Hessel, Matteo, et al. "Rainbow: Combining improvements in deep reinforcement learning." AAAI. (2018).
>
> [2] Henderson, Peter, et al. "Deep Reinforcement Learning That Matters." AAAI. (2018).

---

> > ### Comment · AnonReviewer2 · 2020-11-24
> > **Response to the authors**
> >
> > I appreciate the responses from the authors. Since the authors cannot provide any new experimental results in the revision before the deadline, it is still not clear the proposed method is effective in realistic settings.
> >
> > As for the impact of architecture adaption, again it's not tested on more difficult settings.
> >
> > Therefore, I will keep my original evaluations.

---

> > > ### Author Response · Authors · 2020-11-24
> > > **Response to AnonReviewer2**
> > >
> > > Please excuse the late update of our paper caused by the computational expense of our new experiment. We would like to point you to the recently added experiments in our paper, meta-optimizing Rainbow DQN on Atari environments. Please find details in the recent response to all reviewers. We understand your wish to see results for the latest and perhaps most challenging benchmark of ProcGen. However, due to the time constraints in this rebuttal period, we were unable to meet the computational needs required to perform ProcGen experiments. We hope the provided additional experiments can address your concerns at least to some extent by showing the general usefulness of SEARL to meta-optimize different algorithms in different environments.

---

> > > > ### Comment · AnonReviewer2 · 2020-11-24
> > > > **Response to the authors**
> > > >
> > > > I appreciate the authors' update regarding Rainbow DQN on Atari environments. in Appendix J. It does show that SEARL is more effective than a random search, but other baselines are still missing, which cannot justify the proposed combination of components.
> > > >
> > > > I understand that the authors are working very hard to add more experimental results. But I still don't think the current version is ready for acceptance.

---

> > > > > ### Author Response · Authors · 2020-11-25
> > > > > **Response to AnonReviewer2**
> > > > >
> > > > > We appreciate your thoroughness regarding our evaluation.
> > > > > However, we do want to point out that there is, unfortunately, a lack of comparable AutoRL approaches that tune both the architecture and hyperparameters simultaneously during training the architecture. We provide evidence that SEARL is sample-efficient because of the shared replay memory. Any black-box optimization approach is by design less sample-efficient. What might be less obvious about SEARL is the final agent’s performance. To test this, we compare SEARL to the best randomly sampled configuration of a strong baseline RL algorithm in each respective environment.
> > > > >
> > > > > Further, we want to point out that PBT without modifications cannot serve as a baseline. We were only able to include it in the TD3 case study due to the modifications described. As even the modified PBT variant we used for this case study does not have a mechanism to efficiently use all experiences gathered by the individual agents, we’re confident that SEARL’s advantages w.r.t. sample efficiency will also hold in this case. However, in the final version of the paper, we could provide results from our modified PBT variant in the Arcade Learning environment.
> > > > >
> > > > > Regarding your concerns about the importance of the architecture search, we also want to point out that the experiments in the Arcade Learning Environment suggest that SEARL is capable of navigating even the arguably more complex search space of such vision-based RL tasks. In this context, SEARL adapts both linear and convolutional layers including the channel size, kernel size, and stride.

---

### Official Review · AnonReviewer1 · 2020-10-29

**Rating:** 7
**Confidence:** 5

**Review:**

Motivated by the sensitivity of RL algorithms to the choice of hyperparameters and the data efficiency issue in training RL agents, the authors propose a population-based automated RL framework which can be applied to any off-policy RL algorithms. In the framework, they optimise hyperparameters together with neural architectures. The authors use TD3 on MuJoCo environments as a showcase to demonstrate the advantages of the proposed method. They reduced the number of environment interactions significantly compared to baselines like random search and a modified population-based training algorithm.

Pros:
+ The authors propose an important novel component to PBT-like framework, which is sharing experience among agents in the population. This innovation itself leads to 10x improvement on sample efficiency;
+ Very clear description of the motivations, related work, the details of the training framework, and the experiments. The paper reading has been super pleasant;
+ Results of TD3 on MuJoCo environments, i.e., the ones in Figure 2, are very promising. In the five environments tested, SEARL algorithm shows an order of magnitude improvement on the sample efficiency.

Cons:
- The proposed sharing experience among agents in the population is limited to off-policy RL algorithm, as also noted by the authors;
- There are a few lines of AutoRL and AutoML research are missing in references or in discussions. For example, the architecture search lines of work from Zoph et al. , and for gradient-based meta parameters optimisation like "A Self-Tuning Actor-Critic Algorithm";
- For the main contribution in the paper, shared experience replay within the population, there's a recent work by Schmitt et al. called "Off-Policy Actor-Critic with Shared Experience Replay" which also demonstrates that sharing experience replay in off-policy learning can improve sample efficiency dramatically. Though they are not in the setting of meta learning hyperparameters or architectures, I think it worths discussions;
- From the ablations in Figure 5, it looks like the architecture adaption does not contribute much to the final performance. It might be because MuJoCo environments do not require complex neural architectures. To show that the neural architecture adaptation is a crucial part of the framework (which is a major difference between the proposed method and PBT), the authors might have to move to a complex domain of environments to demonstrate that.

---

> ### Author Response · Authors · 2020-11-18
> **Response to AnonReviewer1**
>
> Thank you very much for the positive rating and the kind words. We appreciate the pointer to recent related work and updated the paper with the following discussions on the mentioned works.
> The work of Zoph et al. [1] on RL for neural architecture search (NAS) is an interesting counterpart to our work on the intersection of RL and NAS: they do RL for NAS (using RL to search better architectures), while we do NAS for RL (changing architectures to improve RL).
> In Schmitt et al. [2], the on-policy experience is mixed with shared experiences across concurrent hyperparameter sweeps to take advantage of parallel exploration. However, this work does not tackle dynamic configuration schedules or architecture adaptation.
> In [3], a subset of differentiable hyperparameters are meta-optimized in an outer loop. This, however, does not extend to non-differentiable hyperparameters and thus does not allow for online tuning of the network architecture. Such hyperparameters can therefore not be meta-optimized in the framework proposed by Zahavy et al.
>
> Impact of architecture adaptation:
> The network architecture has in fact been shown to impact performance in RL before. For example, Henderson et al. [4] experimentally evaluate the impact of architecture choices on MuJoCo. Further, our own ablation study indicates that architecture adaptation is sometimes very beneficial. Specifically, without architecture adaptation, there is a significant performance drop in HalfCheetah and Ant. We updated the paper and included our ablation study from Appendix H in the main paper. We agree with the interest in a more complex domain and further experiments. Therefore we implemented the popular Rainbow-DQN [5] on Atari environments, meta-optimized with the SEARL algorithm. These experiments, however, are very expensive, and we thus do not expect to be able to finish the full 200million frames experiments commonly reported for Atari [5] during the short author response period. But we do believe this would be a valuable experiment and will include it in the final version of the paper.
>
> Thank you again for your valuable remarks!
>
>
>
> [1] Zoph, Barret, and Quoc V. Le. "Neural architecture search with reinforcement learning." ICLR. (2017).
>
> [2] Schmitt, Simon, et al. "Off-policy actor-critic with shared experience replay." arXiv preprint arXiv:1909.11583 (2019).
>
> [3] Zahavy, Tom, et al. "A self-tuning actor-critic algorithm." Advances in Neural Information Processing Systems 33 (2020).
>
> [4] Henderson, Peter, et al. "Deep Reinforcement Learning That Matters." AAAI. (2018).
>
> [5] Hessel, Matteo, et al. "Rainbow: Combining improvements in deep reinforcement learning." AAAI. (2018).

---

> > ### Comment · AnonReviewer1 · 2020-11-24
> > **Response to Authors**
> >
> > I've read through the discussions to added related works. I'm happy with that part, which used to be my concerns.
> >
> > And I'm hoping that we can see results from Rainbow-DQN on Atari games so that we can confirm the proposed method is indeed general and powerful to tune hyperparams in off-policy RL algorithms.

---

> > > ### Author Response · Authors · 2020-11-24
> > > **Response to AnonReviewer1**
> > >
> > > Thank you for your kind response. As an update, we would like to point you to the recently added experiments in our paper, meta-optimizing Rainbow DQN on four Atari environments. Please find details in the recent response to all reviewers. While these experiments are still not ideal in terms of the number of seeds used and environment steps shown, we do hope that they provide you with the necessary insights to evaluate SEARL’s general usefulness to meta-optimize different RL algorithms in different environments.

---

### Official Review · AnonReviewer3 · 2020-10-30
**Official Blind Review**

**Rating:** 5
**Confidence:** 3

**Review:**

Summary: This paper propose a population-based AutoRL framework for hyperparameter optimization of off-policy RL algorithms. The framework optimizes both the hyperparameters and also the neural architecture in a one-shot manner, e.g., search and train at the same time. A shared experience replay buffer is used across the population, which as demonstrated in the experiments, substantially increase the sample efficiency compared to PBT and random search.

Strengths:
- The idea is simple and intuitively makes sense. By sharing the experiences across the population each experience sample gets re-used more often during the training hence the increase in sample efficiency.

Weakness:
- Only apply to off-policy RL.
- Random search is not a very compelling baseline. PBT is not optimized for sample efficiency. It seems that the shared replay buffer can also be applied to PBT? What about differentiable HPO methods? Those are often shown to be much more sample efficient compared to evolution based approaches.
- There is little discussions on what the search results look like, e.g., the learning rate schedule or the neural architecture found by the search method. How are they compared to the SOTA learning rate schedule or neural architecture? Or maybe not even SOTA just compare to some simple heuristics that gradually decrease the learning rate or increase the network size. It would be of limited value if the use of AutoRL can only achieve marginally better performance compared to those simple heuristics.

---

> ### Author Response · Authors · 2020-11-18
> **Response to AnonReviewer3**
>
> We thank the reviewer for the appreciation of our work and the valuable comments.
> We focused on off-policy RL since it is by design generally more efficient due to the usage of a replay memory. In SEARL, off-policy RL can benefit from a diverse population in an evolutionary HPO approach. On-policy remains future work.
> On the baselines:
> Due to the lack of applicable baselines to AutoRL approaches, we use random search (RS) since it is a commonly used HPO technique in RL. In our experiments, RS also acts as a performance baseline for HPO systems rather than a particularly sample-efficient method. The lack of sample-efficiency in PBT is one of the crucial motivations behind SEARL. Adding a shared replay buffer to a PBT-like approach is indeed a crucial part of our contributions, but our approach goes further to explicitly adopt the architecture of the neural networks in the deep RL agent. We added a discussion on recent additional related work as pointed out by reviewer 1, which includes a differentiable HPO approach.
>
> Details of the search results:
> We provided further insights on the learning rate adaptation in Appendix G, and we now use the additional page we have during the author response period to give these insights in the main paper. We now also added a paragraph in the main body, plus a new Appendix I, in which we focus on the found architectures. SEARL finds, e.g., for HalfCheetah larger actor-network sizes (in average 2.8 layers/1000 nodes per network) compared to Walker2d (on average of 1.7layers/680nodes per network). This could indicate that different environments require different network capacities. In such cases, SEARL is capable of automatically adapting the network size to the task complexity.
> Another advantage of the dynamic configuration in SEARL lies in the possibility to achieve on-par performance reported in the TD3 paper with significantly smaller network architectures. Experiments with smaller, static network architectures in TD3 don't achieve comparable performance. This suggests that growing networks like in SEARL is essential for achieving strong performance with small network sizes.
>
>
> Regarding the comparison to simple heuristics:
> We are not aware of any methods for learning rate scheduling in RL or AutoRL which include architecture adaptation (besides evolutionary algorithms we refer to in related work). It is notable that the hyperparameter schedules found by SEARL differ for the individual environments. This could indicate that simple heuristics would also perform well in some environments but not in all. Furthermore, the main goal of SEARL is efficient, automated hyperparameter optimization. The comparison against a handcrafted heuristic is not straight forward, as we cannot easily quantify the computational effort used in creating handcrafted heuristics (it would surely require hyperparameter optimization of its own, so in order to avoid optimizing on the environment of interest (and therefore already paying environment interactions for the tuning stage) one would have to perform leave-one-environment-out hyperparameter optimization. However, in an effort to investigate this valuable point, we ran experiments using simple heuristics to grow the network architecture within the TD3 algorithm over time. As a minimal change, we started using the “add nodes” operator and the same initialization scheme as implemented in SEARL once after the first 200,000 steps in TD3. However, we consistently observe a huge performance drop at this point across all environments, from which TD3 seems unable to recover. This strongly suggests that TD3 itself, without frameworks such as SEARL, is not fit to include simple architecture change heuristics. We suspect that SEARL benefits from the diverse samples in the replay memory to allow more meaningful training updates.
>
> Again, thank you for reviewing!

---

### Official Review · AnonReviewer4 · 2020-10-31
**A good paper with some flaws**

**Rating:** 6
**Confidence:** 4

**Review:**

Summary：
In this paper, the authors intend to propose an efficient automated reinforcement learning (RL) framework. To achieve this goal, they integrate three technologies, i.e., evolutionary RL for hyperparameter search, evolvable neural network for policy network design, and shared experience replay for improving data usage. The paper uses a case study on MuJoCo to demonstrate the claimed advantages over baselines.

Some pros:
1. The motivation is good. The automation of reinforcement learning is beneficial to the research community, especially for researchers who are not familiar with RL but in need of it.
2. The framework is general and friendly to users. This framework is general and can be regarded as a plug-in module for a series of reinforcement learning methods. Besides, as we all know, both of the autoML and RL have a heavy computational burden, the adaption of evolvable neural network and shared experience replay greatly alleviate this dilemma.
3. The organization of this paper is easy to follow. We can follow the authors from why they want to deal with the problem, to how they are inspired by existing work, and then to how the algorithms are design based on the questions to be answered and the existing technologies.

Some cons:
1. The experiment is far from enough. Actually, this paper only has a case study. First, the author claims that the framework can optimize arbitrary off-policy RL algorithms, why only try on TD3? From the perspective of robustness, the authors need to compare more off-policy algorithms with and without the proposed method. Second, the paper claims there is no directly comparable approach for efficient AutoRL, which I do not agree with. In its own related work, many AutoRL baselines are listed, e.g., H. L. Chiang et al.  'Learning navigation behaviors end-to-end with autorl', F. Runge et al. 'Learning to design RNA'. For these baselines, they can either take the same exploration steps and compare performance with the proposed method, or compare the performance/computational cost when reaching the same performance. Anyway, the readers expect to see more comparisons with more baselines from more perspectives. Third, the authors claim that to tackle the non-stationarity of the RL problem, existing studies can substantially increase the number of environment interactions, implying the proposed framework has advantages on non-stationarity RL environments, but still, no experimental results are given.
2. Some technical details are missing. The logic is clear, the solution is reasonable, but the details are ignored. It is a good idea to keep the writing compact, but the lack of details may harm the readability of the paper. For example, since this is a general framework, how should we design hyperparameter settings of SEARL in initialization for different algorithms? In the training part, why should individual be trained for as many steps as frames have been generated in the evaluation phase, and why the training time could be reduced by using only a fraction j of the steps?

---

> ### Author Response · Authors · 2020-11-18
> **Response to AnonReviewer4**
>
> Thank you for your helpful feedback and interest in our paper. We are glad you liked the motivation and clarity of the paper. We fully understand the wish for more experiments on different off-policy algorithms and environments. To this end, we implemented the popular Rainbow-DQN [1] on Atari environments, meta-optimized with the SEARL algorithm. These experiments, however, are very expensive, and we thus do not expect to be able to finish the full 200million frames experiments commonly reported for Atari [1] during the short author response period. But we do believe this would be a valuable experiment and will include it in the final version of the paper.
>
> Concerning the comparison to the related AutoRL work:
> As can be seen in [2], which evaluates [3] on MuJoCo, they do not focus on sample-efficiency in AutoRL: their approach trains thousands of configurations from scratch using individual RL runs. To put SEARL’s sample efficiency in perspective: applying the same fair evaluation protocol as proposed in our paper, it arrives at the same performance with up to 3 orders of magnitude fewer samples on the shown environments (see Figure 1 in evolving rewards paper, scaled by 1000 agents they use). This number is not directly comparable since they use SAC and PPO instead of TD3, but we strongly believe that any differences caused by the choice of algorithm will be far smaller and will not drastically change the sample efficiency achievable with SEARL. We fully agree that we should have made this difference more obvious in our paper and thus point to the fair evaluation protocol already in the related work section.
> ‘Learning to design RNA’ applies BOHB [4] by treating the RL training as black-box and does not focus on online optimization (like PBT or SEARL) nor sample-efficiency. In contrast to a black-box optimization, we jointly train the agent and dynamically optimize the hyperparameters (like PBT). We include random search as a fundamental baseline and to measure performance within the search space.
>
> Addressing the experimental evidence on benefits for the non-stationary nature of RL:
> Our intent by “non-stationary” was focused on the training, e.g. at the beginning of the training some hyperparameter settings are potentially more suitable than others in comparison to later training stages. We see that our wording could be misread and therefore we update the paper accordingly.  Furthermore, in Appendix G we provide details on how the hyperparameters change during the training in different environments. Notably, different environments lead to different hyperparameter schedules (e.g., in HalfCheetah the learning rate decays while in the Hopper and Reacher environment the learning rate even increases). In terms of network architectures, we observe a benefit from a growing architecture in HalfCheetah, whereas the architecture stays small in the Ant environment. We updated the paper with a more detailed discussion on the ablation.
>
> On the missing details:
> We again thank you for your feedback in this regard and updated the paper to describe the technical details more thoroughly.
> We initialize the hyperparameters of the architecture with a small (potentially too small) architecture and keep the learning rate of Adam’s default value. The hyperparameters could also be initialized randomly without large performance differences, see Appendix E. This suggests that there is very little domain-specific knowledge required to effectively use the SEARL framework.
> It is common practice in RL that the agent gets as many update steps as interactions with the environment (we refer to works introducing e.g. TD3, DQN, PPO). This is motivated by the fact that each new observed frame is a new sample that can be used in training. However, we reduce the number of update steps since it would be (A) very expensive since we train a whole population and not a single agent, and (B) since we benefit from the diverse samples gained from the diverse population of agents during our evaluation phase. The more diverse the samples are in the replay buffer, the larger the training gain we can expect from a mini-batch SDG update. This allows us to reduce the number of total update steps per agent.
>
> Again, thank you for your time and effort. We appreciate your feedback.
>
>
> [1] Hessel, Matteo, et al. "Rainbow: Combining improvements in deep reinforcement learning." AAAI. (2018).
>
> [2] Faust, Aleksandra, et al. "Evolving rewards to automate reinforcement learning." AutoML Workshop ICML. (2019).
>
> [3] Chiang, Hao-Tien Lewis, et al. "Learning navigation behaviors end-to-end with autorl." IEEE Robotics and Automation Letters 4.2 (2019).
>
> [4] Falkner, Stefan, et al. "BOHB: Robust and efficient hyperparameter optimization at scale." ICML. (2018).

---

### Author Response · Authors · 2020-11-18
**General Response**

We appreciate the interest in our paper and thank the reviewers for their constructive and valuable feedback. To summarize, we updated our paper-based feedback by:
including the ablation study w.r.t. different SEARL features in the main paper, showing their benefit in individual environments instead of the mean impact;
adding more details about the training process to facilitate reproducibility and improve clarity;
adding more detailed discussions of (additional) related work;
including an additional Appendix section (I), in which we focus on the found architectures by SEARL.
In this updated version, we directly include the appendix as part of the paper for easier navigation. The appendix was previously only part of a zip archive in the supplemental material. In the following, we address each review individually.

We again want to thank all reviewers for their time and effort.

---

### Author Response · Authors · 2020-11-24
**Additional Experiments**

Again, we thank all reviewers for their valuable time and feedback. As mentioned in previous responses, we performed an additional SEARL experiment for meta-optimizing Rainbow DQN on the Atari benchmark suite. We add the paragraph about “Generalization to Different Algorithms and Environments” in Section 4.4 “Results & Discussion” and an additional Section J in the Appendix with a detailed experiment description.

Due to the limited time and the computational budget available to us, we were only able to run experiments for SEARL and random search, with a limited frame count and random seeds. For the final / camera-ready version of this paper, we will increase the number of random seeds, the number of tested Atari environments, as well as environment frames to a similar extent as for the TD3/MuJoCo case study. We will also provide results from our modified PBT approach. However, we can already show similar benefits in terms of sample-efficiency and in performance as observed in the TD3/MuJoCo case study.
We expect this sample-efficiency and performance gain to be present irrespective of the off-policy algorithm being optimized or the environment in which we apply SEARL since the advantages of the proposed meta-optimization in SEARL are generally applicable to any off-policy RL algorithm.
For this reason, we do not expect the findings observed in the current state to change fundamentally with the additional number of seeds/environment steps.
Additionally, all code changes corresponding to additional experiments requested by the reviewers will be updated in the supplemental material.

If we cleared up some of your concerns, we would kindly ask you to update your assessment.

---

### Decision · Program_Chairs · 2021-01-07
**Final Decision**

**Decision:**

Accept (Poster)

**Comment:**

This paper tackles a very important topic in deep RL, namely automatic (non-differentiable) hyper-parameter tuning. It does so by combining ideas from genetic algorithms and neural architecture search with shared experience replay in order to obtain the key property of sample efficiency.  The proposed solution is communicated clearly, and the results are compelling (often 10x improvements), as well as qualitatively interesting.

Unfortunately for the authors, their original submission contained only part of the intended results, hence the borderline scores by some reviewers. In the meanwhile, a second suite of experiments have been added, which I think are compelling enough evidence to validate the paper's approach.